# Systemic Metabolomic Profiles in Adult Patients with Bacterial Sepsis: Characterization of Patient Heterogeneity at the Time of Diagnosis

**DOI:** 10.3390/biom13020223

**Published:** 2023-01-24

**Authors:** Knut Anders Mosevoll, Bent Are Hansen, Ingunn Margareetta Gundersen, Håkon Reikvam, Øyvind Bruserud, Øystein Bruserud, Øystein Wendelbo

**Affiliations:** 1Section for Infectious Diseases, Department of Medicine, Haukeland University Hospital, 5021 Bergen, Norway; 2Section for Infectious Diseases, Department of Clinical Research, University of Bergen, 5021 Bergen, Norway; 3Department of Medicine, Central Hospital for Sogn and Fjordane, 6812 Førde, Norway; 4Section for Hematology, Department of Medicine, Haukeland University Hospital, 5021 Bergen, Norway; 5Leukemia Research Group, Department of Clinical Science, University of Bergen, 5021 Bergen, Norway; 6Department for Anesthesiology and Intensive Care, Haukeland University Hospital, 5021 Bergen, Norway; 7Faculty of Health, VID Specialized University, Ulriksdal 10, 5009 Bergen, Norway

**Keywords:** sepsis, metabolism, metabolomics, bacteremia, organ failure, SOFA score

## Abstract

Sepsis is a dysregulated host response to infection that causes potentially life-threatening organ dysfunction. We investigated the serum metabolomic profile at hospital admission for patients with bacterial sepsis. The study included 60 patients; 35 patients fulfilled the most recent 2016 Sepsis-3 criteria whereas the remaining 25 patients only fulfilled the previous Sepsis-2 criteria and could therefore be classified as having systemic inflammatory response syndrome (SIRS). A total of 1011 identified metabolites were detected in our serum samples. Ninety-seven metabolites differed significantly when comparing Sepsis-3 and Sepsis-2/SIRS patients; 40 of these metabolites constituted a heterogeneous group of amino acid metabolites/peptides. When comparing patients with and without bacteremia, we identified 51 metabolites that differed significantly, including 16 lipid metabolites and 11 amino acid metabolites. Furthermore, 42 metabolites showed a highly significant association with the maximal total Sequential Organ Failure Assessment (SOFA )score during the course of the disease (i.e., Pearson’s correlation test, *p*-value < 0.005, and correlation factor > 0.6); these top-ranked metabolites included 23 amino acid metabolites and a subset of pregnenolone/progestin metabolites. Unsupervised hierarchical clustering analyses based on all 42 top-ranked SOFA correlated metabolites or the subset of 23 top-ranked amino acid metabolites showed that most Sepsis-3 patients differed from Sepsis-2/SIRS patients in their systemic metabolic profile at the time of hospital admission. However, a minority of Sepsis-3 patients showed similarities with the Sepsis-2/SIRS metabolic profile even though several of them showed a high total SOFA score. To conclude, Sepsis-3 patients are heterogeneous with regard to their metabolic profile at the time of hospitalization.

## 1. Introduction

Sepsis is a common disease with high mortality and for many survivors long-term morbidity [1]. It is defined as life-threatening organ dysfunction caused by host responses to various infections, and the organ dysfunction is classified according to the Sequential Organ Failure Assessment (SOFA) score [2,3]. A subset of these patients develops septic shock with profound circulatory/cellular/metabolic dysfunctions and increased mortality; these patients are also characterized by vasopressor requirement and increased serum lactate without hypovolemia [1,2,3]. Furthermore, sepsis is associated with metabolic modulations including mitochondrial dysfunction with altered energy metabolism and production of reactive oxygen species [3], and these dysfunctions are possibly involved in the development of organ failures. Finally, cellular metabolism is also important in the regulation of immunity and inflammation, and modulation of the systemic metabolic regulation can therefore influence inflammatory reactions through effects on the metabolic profile of the microenvironment of cells involved in the regulation of immunity and inflammation [4,5,6,7,8,9,10,11,12,13,14,15]. 

Severe infections often induce an initial acute phase reaction, i.e., a reaction mediated by proinflammatory cytokines and characterized by altered systemic levels of various acute phase proteins (e.g., increased C-reactive protein (CRP) levels, decreased albumin levels) due to the effects of inflammation on distant organs, especially the liver where many of these proteins are synthesized but also their local release at inflammatory sites [16,17]. Sepsis should be regarded as a systemic complication to (at least initially) a local infection involving distant organs [1,2,3]. Sepsis-induced or associated systemic metabolic modulations detected in serum/plasma may thus reflect effects of local infection/inflammation on distant organs similar to the acute phase reaction [16,17,18,19,20], development of various organ dysfunctions [3] and/or the metabolic status/requirements of inflammatory cells at the site of the infection. In this context, we have characterized the metabolomics profiles of patients with bacterial sepsis at the time of hospital admission before the start of any (antibiotic/supportive) treatment. The diagnostic criteria for systemic inflammatory response syndrome (SIRS) and the most recent definition of sepsis (later referred to as Sepsis-3) have been described elsewhere [1,2,3], and in our present study we therefore have a focus on the heterogeneity of SIRS/sepsis patients rather than metabolomic differences between sepsis patients versus healthy controls [20]. 

## 2. Materials and Methods

### 2.1. Patients

The study was approved by the Regional Ethics Committee (REK Vest Norway 214849), and conducted in accordance with the Declaration of Helsinki. Written informed consent for study participation was provided for all patients. The patients received written information about the study when they arrived to the emergency department, and the written informed consent was then signed. For a small minority of severely ill patients, the informed consent was given by the patients’ closest relative/next of kin. Our routines for information and informed consent are consistent with the approval of the Regional Ethics Committee.

Our present study is based on a previous prospective study at Haukeland University Hospital, which is a tertiary hospital in western Norway that also functions as a local emergency hospital for approximately 300,000 inhabitants [21]. Adult patients admitted with sepsis to the emergency department between December 2012 and 2014 were included, and a total of 164 consecutive patients were then admitted with clinical sepsis according to the Sepsis-2 criteria (Appendix A) [22]. However, only 80 of these patients were immunocompetent patients with a later documented bacterial infection and fulfilling the Sepsis-2 criteria at the time of admission [1,2,3]. Patients with viral and parasitic infections, those without proven infections, immunocompromised patients (i.e., known congenital or acquired immunodeficiency), as well as patients receiving immunosuppressive/cytotoxic treatment, were excluded. 

Our present study included 60 patients with bacterial sepsis; 30 patients had infections with Gram-positive and 30 patients with Gram-negative bacteria. We wanted to include 30 patients in each of these two groups to allow reliable bioinformatical comparisons between various patient subsets. Thus, our present study included only 60 out of the 80 patients in the original study that fulfilled the Sepsis-2 SOFA score-based criteria [21]. The 20 excluded patients were:Five patients with mixed infections, i.e., evidence for two infecting bacteria.Eight patients with exceptional bacterial etiology, i.e., one patient each with *Enterobacter cloacae, Acinubaculum schalii, Bacteroides fragilis, Fusobacterium necroforum, Kingella kingae, Neiseria meningitidis, Klebsiella pneumoniae* and *Clostridium* infection.Six (randomly selected from eight) patients for which the bacterial diagnosis was based on detection of bacterial antigens alone but with no bacterial growth for any patient samples.One randomly selected patient with Gram-negative infection (*Escherichia coli* blood culture) that was removed so that we had two equal groups with 30 patients with Gram-negative and Gram-positive infection, separately.

Thus, 19 of the original 80 Sepsis-2 patients were left out to have more homogeneous but still representative patients for the Gram-negative/positive comparison. One additional randomly selected patient was also left out. The remaining 60 patients allowed reliable comparisons of patients with and without bacteremia (30 versus 30 patients) and patients with severe inflammatory response syndrome (25 SIRS or Sepsis-2 patients; see separate description below) versus patients with organ dysfunction/failure (Sepsis-3 patients; 35 of the 60 patients). 

Fifteen patients with Gram-positive and 15 patients with Gram-negative infection included in our present study had bacteremia, and these 30 patients were significantly older than the patients without a blood stream infection (median age 69.5 versus 60 years, Mann Whitney U test, *p* = 0.035). Furthermore, a major difference between patients with Gram-negative and Gram-positive infections was the site of the infection; the majority of patients with Gram-negative infections had urinary tract infections (26/30) whereas patients with Gram-positive infections had mainly respiratory (8/30) or soft tissue infections (11/30). Patients with Gram-negative infections had a higher age (median 73.5 versus 60 years, *p* = 0.043). Finally, our patients with Sepsis-3 and bacteremia included 12 elderly patients with cardiovascular comorbidity. 

The total SOFA score was calculated regularly during the hospital stay, and the information used for this scoring was also documented in the patient journal. In the present study, we refer to the maximal SOFA score documented during the clinical course/hospital stay [23]. Thus, the SOFA score was determined before and independent of the present study.

We classified our patients according to the Sepsis-2 [1] and Sepsis-3 definitions [22,23]. All 60 patients included in the present study fulfilled the Sepsis-2 criteria (i.e., criteria corresponding to SIRS), whereas only 35 patients fulfilled the Sepsis-3 criteria (i.e., based on SOFA definitions). Furthermore, the presence of organ failure at the time of hospital admission according to the Sepsis-3 definition (35 patients out of the 60 patients) and the detection of bacteremia (30 patients) showed a highly significant association (Fisher’s test, *p* = 0.0002). Finally, for the 35 patients with Sepsis-3/organ dysfunction, the subsets with Gram-negative and Gram-positive infections did not differ significantly with regard to their total SOFA score or the frequency of bacteremia (data not shown). The clinical characteristics of Sepsis-3 and Sepsis-2 patients are compared in Appendix A.

The total SOFA scores that are referred to throughout the text, represent the highest score during the hospital stay/treatment period. Our study showed a low mortality with only 3 patients dying within the first four weeks after hospitalization. All three patients fulfilled the Sepsis-3 criteria; and the overall mortality for these patients corresponding to 11.6% is as expected for Sepsis-3 patients [23].

### 2.2. Metabolomic Analyses

Blood samples were derived at the time of admittance to the emergency department of the hospital. Serum was prepared, aliquoted and transferred to storage at −80 °C within 90 min after sampling, and the samples were later stored frozen at −80 °C until analyzed without repeated thawing and freezing of any sample. Thus, we followed highly standardized procedures for sampling, sample preparation and sample storage, but due to the time of sampling at admittance to hospital before any kind of further treatment the samples could not be standardized with regard to food intake or diurnal variations. 

Metabolomic analyses were performed by using the HD4 Analysis Platform of Metabolon (Morrisville, NC, USA). A more detailed description of this methodology is included at the end of the Appendix A. For certain metabolites, certain patients showed undetectable levels; these levels were set to be equal to the lowest detectable level in the statistical and bioinformatical analyses.

### 2.3. Statistical and Bioinformatical Analyses

Following log transformation and imputation of missing values with the minimum observed value for each metabolite, Analysis of Variance (ANOVA) contrast was used to identify metabolites that differed significantly between the various patient subsets. Fisher’s exact test was used for comparison of categorized data, and the Mann-Whitney U-test was used for comparison of continuous data using GraphPad Prism v9.4.1 (Boston, MA 02110, USA). The Pearson’s test was used for correlation analyses. The binomial test for single proportion was also used and Benjamini-Hochberg analysis was used as a correction for the analysis of multiple single metabolites.

We used hierarchical clustering analyses for bioinformatical analysis of patient heterogeneity [21]. Briefly, these were performed using the J-Express software (MolMine AS, Bergen, Norway, http://jexpress.bioinfo.no/site, accessed on 31 November 2022). The mediator values were initially log10 and Z-transformed for standardization of the data before clustering. Our analyses were based on the combination of Euclidean distance and complete linkage because this methodological approach gave the best homology between mediator concentrations and the most compact clusters.

Pathway enrichment analysis displays the number of statistically significantly different metabolites relative to all listed metabolites in a subpathway, compared to the total number of statistically significantly different metabolites relative to all detected metabolites in all other pathways. A pathway enrichment value greater than one indicates that a pathway contains relatively more altered metabolites than all the other pathways together. A list of all metabolic pathways examined (i.e., subgroups) is given in Appendix A. 

## 3. Results

### 3.1. Identified Lipid Metabolites in Serum Samples Derived from Sepsis Patients

We detected a total of 1039 different metabolites in our serum samples; 1011 metabolites were identified but 28 of them were only partially characterized. Detectable levels for at least 54 (90%) of the patients were observed for 660 of these metabolites, and 73 additional metabolites showed detectable levels for at least 48 (80%) of the 60 patients included in the study.

We used principal component analyses (see Appendix A) to analyze our overall results, but these analyses could not be used to separate Sepsis-3/Sepsis-2 patients or patients with and without bacteremia (data not shown).

### 3.2. Comparison of Metabolomic Profiles for Sepsis-3 Patients versus Patients only Fulfilling the Criteria for Sepsis 2: Differences of Amino Acid Metabolite Levels Is a Main Characteristic 

We compared the systemic levels of all detectable metabolites for the 35 patients fulfilling the Sepsis-3 criteria with the levels of the 25 patients only fulfilling the Sepsis-2 criteria. These two patient subsets did not differ significantly with regard to age. 

A total of 97 metabolites showed significant differences between Sepsis-3 and Sepsis-2 patients; the observations are summarized in Table 1 and presented in more detail in Appendix A. For most of these metabolites, we observed a significant correlation between the total SOFA score and corresponding systemic level, but only a small minority of the 97 identified metabolites also showed significant differences when comparing patients with/without bacteremia and patients with Gram-negative versus Gram-positive infections (see Section 3.3 and Section 3.4 below). We detected a total of 1039 metabolites including 227 amino acid metabolites, and this frequency of differing amino acid metabolites (35/227) is significantly different from the frequency of significantly differing metabolites in the other classes (62/805, Fisher’s exact test, *p* = 0.0003). Thus, it is unlikely that the high number of differing amino acid metabolites is caused by coincidence. 

The large majority of the 97 metabolites showed detectable levels for most patients; 72 metabolites reached detectable levels for at least 54 patients (>90% of the patients) and nine additional metabolites reached detectable levels for at least 48 patients (>80%). Low levels were detected especially for xenobiotic metabolites; 12 of the 16 metabolites with detectable levels in less than 48 patients were classified as xenobiotic/drug metabolites. 

We observed significant differences for 40 heterogeneous amino acid metabolites and peptides, and the large majority of these metabolites showed increased levels in Sepsis-3 patients. The largest groups were histidine and leucine/isoleucine/valine metabolites (six metabolites in each group), but five Urea cycle; Arginine and Proline Metabolism metabolites were also significantly altered. Furthermore, xenobiotics constituted an additional large group, including six food components and also 11 drug metabolites reflecting the cardiovascular comorbidity for a subset of our Sepsis-3 patients. 

We did an additional pathway enrichment analysis, and even though amino acid metabolites constituted the largest group of differing metabolites (Figure 1), additional differences of potential biological importance were also identified:Several terms reflecting altered amino acid metabolism also received a high score in this analysis, including a specific amino acid metabolism, as well as the urea cycle and polyamine metabolism.The differences in fibrinogen cleavage peptides possibly reflect acute inflammation and the acute phase reaction.There were differences in fatty acid metabolism (acyl cholines, eicosanoid).There was a difference in several xenobiotics and especially cardiovascular drugs; this reflects the cardiovascular comorbidity of a Sepsis-3 patient subset.There were also differences in pentose/aminosugar metabolism and glycolysis/gluconeogenesis/pyruvate metabolism suggesting that the carbohydrate metabolism is altered together with the amino acid metabolism.Vitamin and cofactor metabolism showed differences in nicotinate and nicotinamide metabolism as well as ascorbate and aldarate metabolism.

Taken together, the pathway enrichment analysis suggests that Sepsis-3 patients show complex metabolic differences when compared with Sepsis-2/SIRS patients.

### 3.3. Comparison of Metabolomic Profiles for Sepsis Patients with and without Bacteremia: Differences in Systemic Levels of Lipid and Amino Acid Metabolites Are Main Characteristics 

As described in Section 1, the detection of bacteremia was significantly associated with a relatively high total SOFA score; bacteremia should therefore be regarded as a sign of more severe disease. For this reason, we also did a comparison of the metabolic profile for the patients with and without bacteremia (30 patients in each of these two groups, see Appendix A). These two patient groups/subsets differed significantly with regard to age; patients with bacteremia were significantly older (median age 69.5 years, range 32–96 years) than patients without bacteremia (median 60 years, range 20–84 years, Mann-Whitney U-test, *p* = 0.035).

The results from the comparison of the 30 patients with and the 30 patients without bacteremia are presented in detail in Appendix A and summarized in Table 2. We identified a total of 51 heterogeneous metabolites, the two largest groups were 11 amino acid metabolites and 16 lipid metabolites. However, the 51 metabolites also included nine xenobiotic metabolites that reflected the patients’ cardiovascular comorbidity, whereas the remaining 42 metabolites probably reflect the effects of sepsis/bacteremia on the systemic metabolomics profile. Only 10 of these 42 sepsis-modulated metabolites showed increased levels in patients without bacteremia, and this is significantly different from the equal distribution (i.e., 21 increased and 21 decreased) that would have been expected if these 42 metabolic differences were caused by coincidence alone (binomial probability test, *p* = 0.0047).

The large majority of the 51 metabolites showed detectable levels for most of the 60 patients included in the study; 30 metabolites reached detectable levels for at least 54 patients and six additional metabolites reached detectable levels for at least 48 patients. Low levels were detected especially for xenobiotic metabolites; 10 of the 15 metabolites with detectable levels in less than 48 patients were xenobiotic/drug metabolites. 

We did a pathway enrichment analysis based on the 51 metabolites showing significantly different levels when comparing patients with and without bacteremia (Figure 2). Some amino acid subpathways showed a high score (methionine, cysteine, taurine). Two peptide subclasses (fibrinogen cleavage peptides, gamma-glutamyl amino acid) also showed factors exceeding 2. The lipid metabolites were heterogeneous (Table 2), and altered levels were especially seen for corticosteroids and progestin metabolites, but also for eicosanoids and several fatty acid metabolic pathways. These relatively high scores for a limited number of subclasses cannot be explained by coincidence alone.

### 3.4. Comparison of Metabolomic Profiles for Patients with Gram-Negative and Gram-Positive Infections: Only Weak Associations Are Detected 

Several molecules derived from Gram-positive and Gram-negative bacteria differ in their binding to pattern-recognizing receptors [24,25,26], and such differences may then lead to differences with regard to the function (possibly including the metabolic regulation) of receptor-expressing cells. For this reason, we compared the metabolomics profiles for patients with Gram-negative and Gram-positive infections and identified 39 metabolites that differed significantly between these two patient subsets; 17 of them were xenobiotic metabolites (mainly drug metabolites) and the other 22 biochemicals represent a heterogeneous group of various endogenous metabolites (Appendix A). Patients with Gram-negative infections were significantly older than the patients with Gram-positive infections (median age 73.5 versus 61 years, Mann-Whitney U-test, *p* = 0.43), and the Gram-negative patients also included a higher frequency of patients with urinary tract infections (2 versus 26 patients, *p* < 0.00001). The frequencies of respiratory (1 versus 8 patients, *p* = 0.0257) and soft tissue infections (none versus 11 patients, *p* = 0.0003) were significantly lower; a SOFA score ≥2 was less common (13 versus 22 patients, *p* = 0.018) and respiratory failure was more common (12 versus 23 patients, *p* = 0.004) for patients with Gram-positive infections.

An additional pathway enrichment analysis showed relative weak associations only for a few pathways. Only four pathways showed an enrichment corresponding to a factor exceeding 2, and all four pathways showed only a borderline enrichment with progestin steroids 2.18, pregnenolone steroids 2.18, fatty acid metabolism (acyl carnitine, dicarboxylate) 2.17 and aminosugar metabolism 2.17.

### 3.5. Metabolic Heterogeneity of Sepsis Patients: A Clustering Analysis Based on Amino Acid Metabolites Showing a Strong Correlation with the Total SOFA Score 

A total of 450 metabolites showed significant correlations with the total SOFA score (i.e., *p-*value < 0.05, Pearson’s test) for our 60 patients, and many of the metabolites that differed significantly when comparing Sepsis-3 versus Sepsis-2 patients were also amino acid metabolites which showed a correlation with the total SOFA score. To further elucidate the associations between metabolomic status and disease severity/organ dysfunction, we identified and characterized those metabolites that showed a particularly strong correlation with the total SOFA score. We first identified all metabolites that showed a strong association with total SOFA score, i.e., individual metabolites that showed a *p*-value < 0.005 and correlation corresponding to >0.6 (Appendix A). We identified a total of 42 metabolites that fulfilled both these criteria, and all these metabolites showed very low *p*-values that were still regarded as significant after Benjamini-Hochberg evaluation for multiple testing. Furthermore, the large majority of the 42 metabolites showed detectable levels for most patients; only eight metabolites (including four amino acid metabolites) showed detectable levels in less than 54 of the patients included in the study. 

We did an unsupervised hierarchical clustering analysis based only on the 23 amino acid metabolites included among these 42 highly significant metabolites (Figure 3, Appendix A). Based on this analysis, we identified a lower main cluster/subset including six patients, whereas the upper main patient cluster including the remaining 54 patients could be further divided into two subclusters that included 26 (upper subcluster) and 28 patients (lower subcluster), respectively. We then compared (i) the 26 patients in the upper subcluster (i.e., the upper 26 patients in the clustering Figure 3) with (ii) the 34 patients in the lower subcluster plus the six patients in the lower main cluster (i.e., the lower 34 patients in the clustering Figure 3) (Table 3). The sample storage time did not differ between these two patient subsets. The 34 lower patients showed an increased frequency of Sepsis-3 patients, an increased SOFA score, higher age and increased serum creatinine levels, but the two patient subsets did not differ with regard to peripheral blood cell counts (total leukocytes/neutrophils/monocytes/platelets), blood pressure (systolic/diastolic), respiratory rate or the frequencies of individual organ failures as defined by the Sepsis-3 criteria at the time of sampling (data not shown). Finally, all patients with cardiovascular comorbidity had Sepsis-3 and were included among the lower 34 patients. Thus, most Sepsis-3 patients show an amino acid profile that differs from Sepsis-2 patients, but Sepsis-3 patients are heterogeneous and a subset of them shows a profile similar to many patients who only fulfill the Sepsis-2 criteria.

We also compared the eight Sepsis-3 patients included in the upper subcluster (i.e., among the upper 26 patients) with the 27 Sepsis-3 patients included among the other lower 34 patients (i.e., lower subcluster plus lower main cluster) (Table 4, Figure 3). We emphasize that these data should be interpreted with great care because the upper subcluster included a low number of Sepsis-3 patients. First, the two patient subsets did not differ significantly with regard to the total SOFA score. Second, the age difference reached only borderline significance, whereas the creatinine level was significantly higher for the lower 27 patients. Third, these two Sepsis-3 patient subsets did not differ with regard to peripheral blood cell counts (total leukocytes/neutrophils/monocytes/platelets), blood pressure (systolic/diastolic), respiratory rate or the frequencies of individual organ failures as defined by the Sepsis-3 criteria (data not shown). Finally, all patients with cardiovascular comorbidity were included among the lower 34 Sepsis-3 patients.

### 3.6. A Minor Subset of Lipid Metabolites Show a Strong Correlation with Total SOFA Score

Although the majority of the 42 metabolites showing a strong association with the total SOFA score were classified as amino acid metabolites (23 metabolites), a minor subset of 12 lipid metabolites also showed a similar strong correlation with the total SOFA score, i.e., *p*-value < 0.005 and a correlation factor > 0.6 (Appendix A). We did an unsupervised hierarchical cluster analysis including all 60 patients based on these 12 metabolites (Appendix A). This analysis identified two main patient clusters; the upper cluster included 17 Sepsis-3 patients among a total of 37 patients, whereas the lower main cluster included 18 Sepsis-3 patients among 23 patients (Fisher’s exact test, *p* = 0.0168). Even though these lipid metabolites also identified a patient subset with a Sepsis-3 associated metabolic profile, this difference between Sepsis-2 and Sepsis-3 patients was less significant than the subclassification based on clustering of the amino acid metabolites (Section 3.4, Figure 3). Thus, these observations further support the hypothesis that the development of organ failure in patients with severe bacterial infections (i.e., Sepsis-3 patients) mainly alters the systemic levels of amino acid metabolites.

The lipid metabolites that showed a strong correlation with the maximal total SOFA score (i.e., *p*-value < 0.005, correlation factor > 0.6) included mainly pregnenolone and progestin metabolites. We, therefore, compared the correlations with the maximal total SOFA score for all 57 steroid hormone metabolites that were detected in our 60 patients (only one estrogen metabolite analyzed) (Appendix A). It can be seen that the 16 pregnenolone/progestin metabolites differed from the other 41 steroid hormone metabolites and showed relatively stronger associations with the total SOFA score (correlation > 0.5 for 11/16 metabolites and >0.4 for all 16 metabolites) compared with the other 41 steroid hormone metabolites (correlation > 0.5 1/41; Fisher’s exact test, *p* < 0.00005). Finally, Sepsis-3 patients included a lower number of female patients compared with patients only fulfilling the Sepsis-2 criteria (13/25 versus 16/35), but this difference did not reach statistical significance (*p* = 0.0659). 

### 3.7. The Heterogeneity of Sepsis Patients Characterized by Unsupervised Hierarchical Clustering Analysis Based on Heterogeneous Metabolites Showing Strong Association with Total SOFA Score

We identified 42 heterogeneous metabolites (including the 23 amino acid and 12 lipid metabolites described above) that showed a strong association with the total SOFA score (Appendix A), i.e., *p* < 0.005 and correlation factor > 0.6 in the Pearson test. All these metabolites showed a low *p*-value that remained significant after Benjamini-Hochberg correction, but they represent only a minority of the metabolites that showed a *p*-value below 0.05 (statistical significance) and a correlation value > 0.5 when testing for correlation with the total SOFA score (Appendix A). 

We did a clustering analysis based on the 42 highest ranked metabolites. This analysis included only metabolites that reached detectable levels for at least 10 patients; the results are shown in Figure 4 and the metabolite clustering is presented in detail in Appendix A. Most Sepsis-3 patients clustered close to each other also in this analysis. This is illustrated by our classification of patients in two main subsets by separating them between two patient subclusters; we then define an upper subset including 28 patients and another subset including the 32 lower patients (Figure 4). These two patient subsets differed significantly in the frequency of Sepsis-3 patients (8/28 in the upper and 27/32 in the lower subgroup, Fisher’s test, *p* = 0.0033). However, the definition of two patient subsets was less obvious for this 42 metabolite analysis than for the 23 amino acid clustering (Figure 3). 

The clustering analyses presented in Figure 3 and Figure 4 showed that some exceptional Sepsis-3 patients had a metabolomic profile similar to Sepsis-2/SIRS patients. Notably, a considerable overlap was obsrrved between the exceptional patients identified in clustering analyses based on high-ranked amino acid metabolites and the 42 highest ranked metabolites. The exceptional metabolites from the amino acid clustering were either included among the exceptional patients or close in their neighboring cluster in the analysis based on all 42 highest-ranked metabolites. 

The clustering of the 42 metabolites is also presented in Appendix A and the following conclusions can then be made from this overview:Only a minority of our sepsis patients showed increased levels of most of the 42 metabolites; these patients clustered close to each other and represent the eight patients in the lowest part of the cluster analysis.All the other 52 patients showed relatively low levels for several of the 42 metabolites, and especially low levels of metabolites belonging to metabolite cluster B (middle left; see Appendix A), which seems to be a common characteristic both for the other Sepsis-3 patients and patients only fulfilling the Sepsis-2 criteria. Cluster B is a small and heterogeneous cluster including seven metabolites (four of them being amino acid metabolites).The patients in the upper part of the diagram also showed generally low levels for metabolite clusters A and C. Cluster A (left) is a relatively large and heterogeneous metabolite cluster whereas cluster C (middle right) includes 15 metabolites and 12 of them are amino acid metabolites.Cluster D (right) includes only lipid metabolites and decreased levels were seen, especially for the upper 12 patients.

We finally did a clustering based on 131 total SOFA-associated metabolites with *p*-values < 0.01 and correlation factor > 0.5. This analysis also identified a small minority of Sepsis-3 patients with generally high metabolite levels, a heterogeneity of Sepsis-3, with a majority of them showing a metabolic profile that was different from most patients who only fulfilled the Sepsis-2 criteria, and a minority of Sepsis-3 patients who showed a metabolic profile similar to most Sepsis-2/SIRS patients (data not shown).

## 4. Discussion

In our present study, we conducted a broad metabolic characterization of patients with Sepsis based on the Sepsis-2 and Sepsis-3 criteria. The study was based on a previous clinical study of consecutive patients, and samples were derived when patients were admitted to the hospital. We describe that most patients with Sepsis-3 (i.e., with organ failure) have a systemic metabolic profile at admittance which differs from Sepsis-2/SIRS patients. However, Sepsis-3 patients are heterogeneous with regard to their metabolic profile, especially with regard to amino acid metabolites. 

Patients with sepsis are very heterogeneous with regard to their clinical characteristics. First, many patients develop sepsis as a relatively late complication following trauma and/or surgical treatment [2,3,27,28,29]. Second, many of the sepsis patients also have severe comorbidity or they show abnormalities in their immunocompetent/inflammatory cells due to frailty and/or aging [30,31,32,33,34,35]. Third, sepsis can be caused by a wide range of various microorganisms, including both bacteria and fungi, and bacterial molecules may then interact with and modulate the functions of various immunocompetent cells [24,25,26]. Finally, the infection site may vary, and endothelial cell differences between various vascular beds and/or the interactions between infection site cells and infiltrating immunocompetent/inflammatory cells in their common microenvironment may then differ between various sites [36]. For these reasons, we investigated a well-characterized and relatively homogenous group of sepsis patients, i.e., patients admitted to the emergency department of a general hospital with sepsis caused by common bacterial agents and handled without surgical interventions. The patients showed a (relatively limited) clinical heterogeneity, but even early in their disease course they showed an additional metabolic heterogeneity, and even exceptional patients with a high total SOFA score showed a systemic metabolic profile similar to Sepsis-2/SIRS patients. 

Our Sepsis-3 patients showed increased lactate levels, and their overall early mortality was 11.7% (i.e., three Sepsis-3 patients). These observations would be expected for Sepsis-3 patients, and they suggest that our 60 patients as well as our Sepsis-3 patients are representative. However, due to this low mortality, detailed survival analyses were not possible in our present study. 

As described in the Results section, some of the metabolites showed undetectable levels for certain patients, and in the statistical and bioinformatical analyses undetectable levels were set equal to the lowest detectable level. This was done to avoid overestimation of differences between compared groups. However, for the majority of metabolites, detectable levels were observed for more than 90% of the patients, and this was also true for the large majority of metabolites that differed significantly between Sepsis-3 and Sepsis-2 patients, between patients with and without bacteremia and for metabolites showing strong associations with the total SOFA score.

We first compared patients with Sepsis-3 with patients who only fulfilled the Sepsis-2 criteria. Several metabolites showed significant differences, and amino acid metabolites constituted the largest group of differing metabolites. Our Sepsis-3 patients also showed a significant increase in several cardiovascular drug metabolites; these metabolites generally showed no associations with the total SOFA score, and the increase in these metabolites seem to reflect inclusion of a subset of elderly patients with Sepsis-3, bacteremia and cardiovascular comorbidity. Furthermore, we also did a second comparison of patients with and without bacteremia. We then detected fewer metabolites that differed significantly between these two groups; the differing metabolites included a larger subset of lipid metabolites but again we observed increased levels of several cardiovascular drug metabolites. 

Our two comparisons of (i) patients with Sepsis-3 versus Sepsis-2 and (ii) patients with/without bacteremia showed only a minor overlap of the identified differing metabolites, e.g., the sepsis comparison identified a large group of amino acid metabolites whereas the bacteremia comparison identified several lipid metabolites. However, it should be emphasized that both comparisons identified mainly metabolites that showed significant correlations with the maximal total SOFA score, but with the exception of most of the identified xenobiotic metabolites that did not show significant correlations with SOFA score/disease severity. In our opinion, these metabolites reflect age/comorbidity rather than sepsis-induced metabolic modulation. This hypothesis is also consistent with our observation that very few xenobiotic metabolites were included among the highest-ranked metabolites with regard to correlation with the SOFA score. 

We investigated correlations between metabolite serum levels at the time of admission and total SOFA score. A large subset of metabolites showed significant associations with the SOFA score, but we focused on the highest-ranked metabolites and these metabolites also included a large number of amino acid metabolites. For these reasons, we have a focus on high-ranked amino acid metabolites in our clustering analyses of metabolic profiles. 

Modulation of systemic amino acid profiles are associated with prognosis in patients with various forms of cancer [37,38,39,40,41,42]; and in patients with colorectal cancer decreased levels of glutamine and histidine together with increased levels of phenylalanine are associated with systemic signs of cancer-associated inflammation [37]. Thus, amino acids and/or amino acid profiles are regarded as potentially useful biomarkers in several malignant diseases, and our present results suggest that amino acid (profiles) should be further investigated as possible prognostic markers also in patients with other inflammatory diseases, including bacterial sepsis. 

Our study was based on a previous clinical study including consecutive adult patients requiring emergency admission to hospital due to suspected sepsis [21]. All patients had bacterial infections and were relatively homogeneous with regard to bacterial etiology, and only four of them had a predisposition for sepsis that required surgical evaluation. In our opinion, our present study is therefore representative of medical patients with sepsis. However, sepsis patients are heterogeneous [1,21,43], and our study may not be representative for patients with uncommon bacterial etiologies, sepsis in patients requiring surgical intervention, sepsis after surgery or trauma, pediatric sepsis or sepsis in immunocompromised patients. Finally, additional studies are also needed to clarify the possible impact of the infection site on metabolomic profiles, although our observation that relatively few and heterogeneous metabolites that differed between patients with Gram-negative (many urinary tract infections) and Gram-positive infections (very few urinary tract infections) suggests that the impact of the infection site on the metabolomic profiles may be limited. 

Both Gram-negative and Gram-positive bacteria express molecules that show specific binding to various receptors expressed by immunocompetent cells as well as other cells involved in the regulation of inflammation, e.g., lipopolysaccharide that binds to toll-like receptor (TLR)4 and lipoteicoic acid that binds to TLR2 [24,25,26,44,45,46,47,48,49,50,51,52,53,54]. Thus, bacteria-derived molecules can alter the function of inflammatory cells, probably including modulation of their metabolism [43]. Despite the differences between with Gram-positive and Gram-negative infections with regard to ligation of pattern-recognizing receptors, these two patient subsets showed relatively small differences in their systemic metabolomic profiles. The explanation for this is probably that organ failure has a much stronger impact on the systemic metabolic regulation than the bacterial etiology. 

Our clustering analyses identified patient subsets with Sepsis-3 associated and Sepsis-2/SIRS associated metabolic profiles, and these patient subsets also differed with regard to renal function/creatinine levels (Table 3 and Table 4). The association between metabolic profile and creatinine level was also observed when Sepsis-3 patients were investigated alone (Table 4). Renal failure is associated with altered levels of a wide range of metabolites, including altered amino acid metabolism [55,56,57,58,59]. The metabolic differences described in our present study, including altered amino acid metabolism, may therefore at least partly be caused by altered renal function.

Cellular reprogramming of metabolism is regarded as one of the mechanisms behind the organ failure in patients with sepsis, but the effect of targeting these mechanisms will possibly depend on the timing of such therapeutic interventions [60]. Our present study describes the metabolic status of Sepsis-3 patients at the time of hospital admission/diagnosis and may therefore contribute to a scientific basis for the design of early interventions that target metabolic mechanisms behind the development of sepsis-associated organ failure.

Several previous studies comparing the metabolic profiles between sepsis patients and healthy controls have described increased levels of amino acids or amino acid metabolites in sepsis patients together with increased levels of metabolites reflecting mitochondrial functions/energy metabolism (e.g., lactate) and altered regulation/function of the urea cycle [43,60]. It should be emphasized that these different metabolic systems are interacting as described and visualized by Hussain et al. [43]. Our study shows that several of these markers, in addition, differ between Sepsis-3 and Sepsis-2 (SIRS) patients. However, several amino acid metabolites showed a strong correlation with the maximal total SOFA score, and this observation suggests that the early abnormalities in amino acid metabolism detected at hospital admission are not only early markers of sepsis but also reflect the later clinical course/maximal total SOFA score.

Therapeutic targeting of metabolism is now considered as a possible therapeutic strategy in sepsis [61]. Animal studies suggest that metabolic intervention can decrease the mortality in sepsis [62,63,64]; the same is suggested by subgroup analyses for some studies in humans [43,61] but the results of randomized clinical studies of metabolic interventions in sepsis have generally shown no significant improvement of patient survival by this therapeutic strategy [61]. However, sepsis patients show a considerable clinical heterogeneity [1], and our present study clearly illustrates that sepsis patients also show a metabolic heterogeneity even when investigating a relatively homogeneous (but still representative/consecutive) group of patients admitted to the emergency unit of a medical department. The metabolic heterogeneity described in our present study represents a difference in the metabolic context of therapeutic interventions, and such differences may the influence the effects of therapeutic targeting of systemic metabolic regulation. 

Our present study was based on a previous clinical study including consecutive adult patients with suspected severe infections requiring emergency admission to hospital [21]. All patients had bacterial infections; the investigated group showed a limited heterogeneity with regard to bacterial etiology, and only four of them had a predisposition that required surgical evaluation. Our study is thus representative of medical patients with sepsis. However, it should be emphasized that our results may not be representative of sepsis patients with uncommon bacterial etiologies or fungal infections, sepsis in patients requiring or following surgical interventions, sepsis as a complication after trauma, pediatric sepsis or sepsis in immunocompromised patients. Future studies have to clarify whether our present results are representative also for (any of) these patient groups. There is also a need to further investigate the possible impact of the infection site on systemic metabolic profiles in sepsis, but the relatively few and heterogeneous metabolites that differed between patients with Gram-negative (many urinary tract infections) and Gram-positive infections (very few urinary tract infections) suggest that the impact of the infection site may be limited. 

In a recent study, we compared the lipidomic profiles of Sepsis-3 and Sepsis-2 patients [65]. We could not detect any extensive lipidomic differences between these two groups, although limited differences for certain subsets of lipid metabolites were observed, especially for lysophosphatidylcholines and sphingolipids. In the present study, we observed that Sepsis-3 and Sepsis-2 patients mainly differ with regard to amino acid metabolites, and several amino acid metabolites also showed strong correlations with the total SOFA score. Thus, when using a different methodological approach in the present study we confirmed that Sepsis-3 and Sepsis-2 patients do not show extensive differences with regard to their lipidomic profiles.

A minority of pregnenolone/progestin metabolites showed a strong correlation with the maximal total SOFA score, and when analyzing the overall results for steroid hormone metabolites (Appendix A), pregnenolone/progestin metabolites showed generally stronger associations with total the SOFA score than other steroid hormone metabolites. This is probably due to stress/sepsis-induced modulation of steroid/cortisol metabolism as described in several previous studies [66,67,68]. Sex-associated differences are unlikely as an explanation for these pregnenolone/progestin effects because Sepsi-3 and Sepsis-2 patients did not differ significantly with regard to male/female distribution; most of these metabolites reached detectable levels both for males and females, and an inverse effect was not observed for androgens. Thus, altered pregnenolone/progestin metabolic profiles seem to be an early marker for the later severe clinical course of patients with bacterial sepsis. 

## 5. Conclusions

Our present study shows that most Sepsis-3 patients have a different serum metabolomic profile at the time of hospital admittance when compared with only Sepsis-2/SIRS patients (i.e., without organ failure), especially with regard to amino acid metabolism. Several amino acid metabolites are also among the highest-ranked metabolites with regard to association with the total SOFA score. However, our analyses showed that Sepsis-3 patients are heterogeneous with regard to their systemic metabolic profiles at the time of hospital admission. Metabolic targeting to reduce the risk of severe sepsis-associated organ failure has been suggested [61]. Our present study suggests that sepsis patients are heterogeneous with regard to metabolic regulation; and for this reason, one should possibly consider individualizing this treatment if metabolic targeting is tried early after hospital admission. 

## Figures and Tables

**Figure 1 biomolecules-13-00223-f001:**
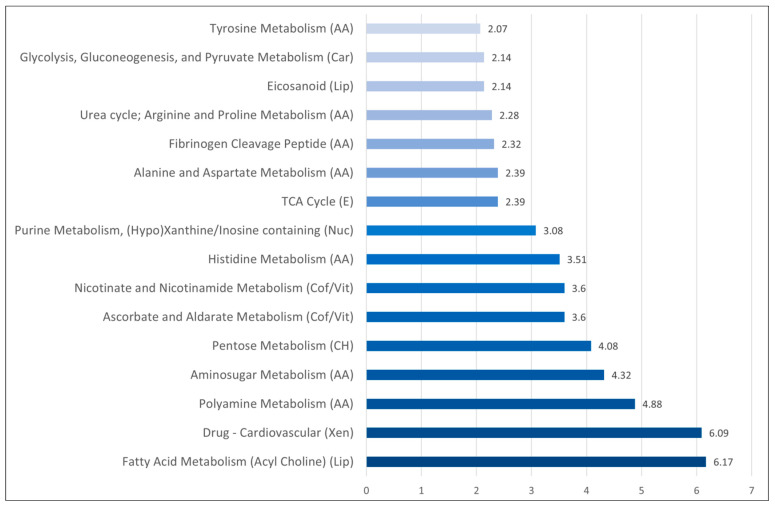
Pathway enrichment analysis based on the 97 metabolites that showed significant differences when comparing the 35 patients fulfilling the Sepsis-3 criteria versus the 25 patients only fulfilling the Sepsis-2 criteria. The abscissa axis refers to the pathway enrichment factor that is defined in Section 2.3. (Abbreviations: AA, amino acid; Car, carbohydrate metabolism; CH, carbohydrate pentose metabolism; Cof/Vit, cofactors and vitamins; E, energy; Lip, lipid; Pep, peptide; Nuc, nucleotide; Xen, xenobiotics).

**Figure 2 biomolecules-13-00223-f002:**
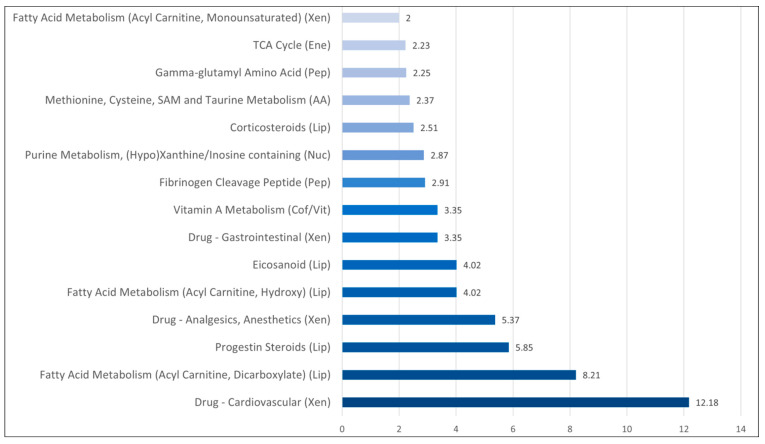
Pathway enrichment analysis based on the 51 metabolites that showed significant differences when comparing patients with and patients without bacteremia. The abscissa axis refers to the pathway enrichment factor that is defined in Section 2.3. (Abbreviations: AA, amino acid; Cof/Vit, cofactors and vitamins; Ene, energy; Lip, lipids; Pep, peptide; Nuc, nucleotide; Xen, xenobiotics).

**Figure 3 biomolecules-13-00223-f003:**
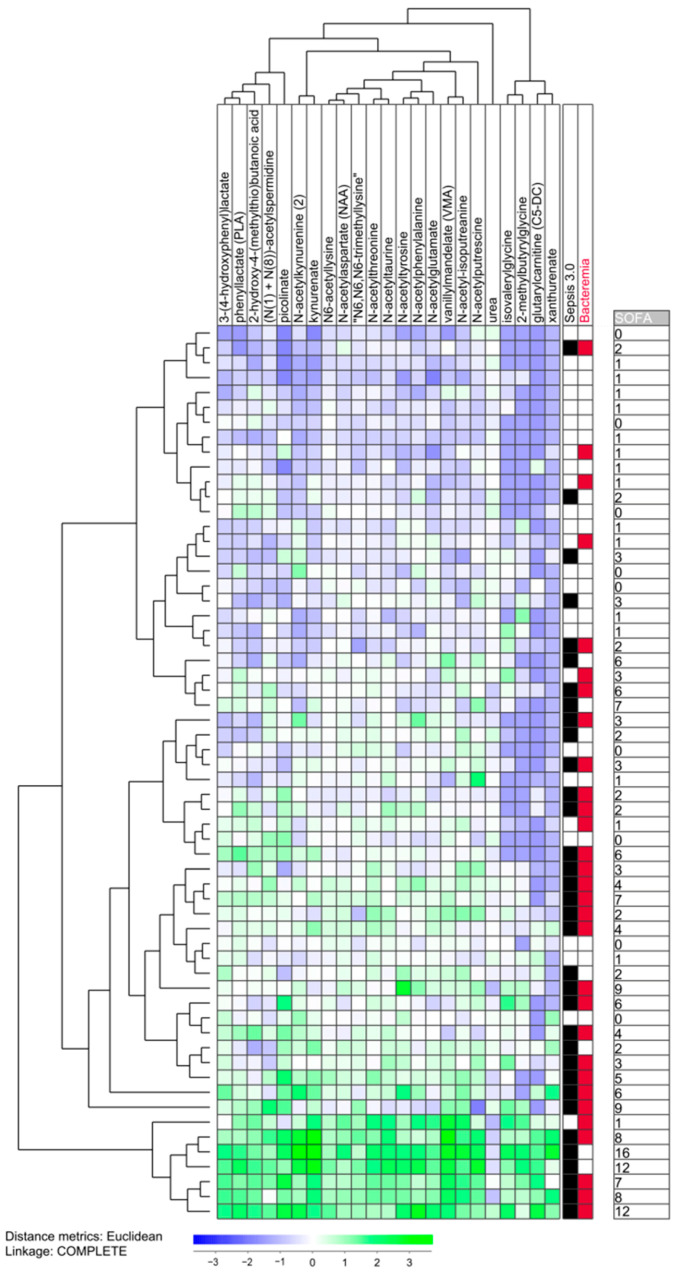
Subclassification of sepsis patients based on amino acid metabolites that showed a strong correlation with the total SOFA score. We performed an unsupervised hierarchical clustering analysis based on 23 amino acid metabolites that showed a strong correlation with total SOFA, i.e., Pearson’s correlation test with *p*-value < 0.005 and a correlation factor > 0.6. All metabolites showed detectable levels for at least 10 patients The characteristics of each individual patient (fulfilling Sepsis-3 criteria (black color/blocks), detection of bacteremia (red color/blocks) and total SOFA score) are indicated to the right in the figure.

**Figure 4 biomolecules-13-00223-f004:**
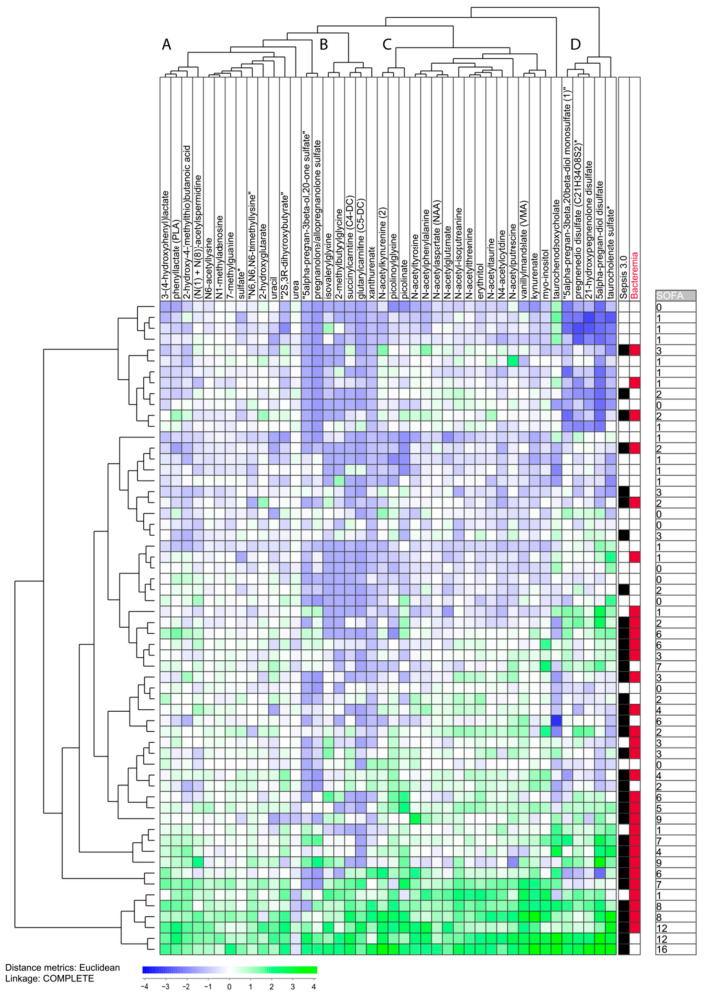
Subclassification of sepsis patients based on 42 top-ranked individual metabolites that all show a strong correlation with the total SOFA score, i.e., *p* < 0.005 and correlation factor > 0.6 with the Pearson’s test. All metabolites showed detectable levels for at least 10 patients. The characteristics of each individual patient are indicated to the right in the figure: Patients fulfilling Sepsis-3 criteria (black blocks), detection of bacteremia (red blocks) and total SOFA score (value). The metabolite clusters are referred to as clusters A–D (see also Appendix A). * The metabolite identity not confirmed based on a standard.

**Table 1 biomolecules-13-00223-t001:** Differences in individual metabolites when comparing patients fulfilling the Sepsis-3 definition versus patients only fulfilling the Sepsis-2 criteria.

Subpathway	Identity
AMINO ACIDS AND PEPTIDES (40/228 metabolites altered, 31 correlated with SOFA score)	
Alanine and Aspartate Metabolism (2/9)	*N*-acetylalanine, *N*-carbamoylalanine
Glutamate Metabolism (1/12)	carboxyethyl-GABA
Histidine Metabolism (6/19)	1-methylhistidine, *N*-acetyl-1-methylhistidine *, imidazole lactate, 1-methyl-4-imidazoleacetate, 1-methyl-5-imidazoleacetate, 1-methyl-5-imidazolelactate
Lysine Metabolism (1/19)	5-(galactosylhydroxy)-lysine
Tyrosine Metabolism (4/)	*N*-acetyltyrosine, 3-(4-hydroxyphenyl)lactate, phenol sulfate, *N*-formylphenylalanine,
Tryptophan Metabolism (2/25)	Kynurenine, indolelactate
Leucine, Isoleucine and Valine Metabolism (6/27)	beta-hydroxyisovaleroylcarnitine, 3-methylglutaconate, Isoleucine, tiglylcarnitine (C5:1-DC), 3-hydroxy-2-ethylpropionate, *N*-carbamoylvaline
Methionine, Cysteine, SAM and Taurine Metabolism (3/26)	cysteine, *N*-acetylmethionine sulfoxide, 2,3-dihydroxy-5-methylthio-4-pentenoate (DMTPA) *
Urea cycle; Arginine and Proline Metabolism (5/24)	Arginine, urea, homocitrulline, *N*2*,N*5-diacetylornithine, argininate
Polyamine Metabolism (4/9)	*N*(1) *+ N*(8))-acetylspermidine, Acisoga, *N*1*,N*12-diacetylspermine, 4-acetamidobutanoate
Guanidino and Acetamido Metabolism (1/4)	guanidinosuccinate
Peptides (5/50)	gamma-glutamylserine, isoleucylglycine, fibrinopeptide A (3–15) *, fibrinopeptide B (1–11) * fibrinopeptide B (1–12) *
CARBOHYDRATE AND ENERGY (all increased, all associated with the SOFA score)	
Carbohydrate (6/76)	Lactate, ribonate, arabinose, arabitol/xylitol, galactonate, erythronate *, *N*-acetylglucosamine/*N*-acetylgalactosamine
Energy metabolism (2/10)	alpha-ketoglutarate, fumarate
LIPIDS (7 increased including the acyl cholines, 8 associated with thevSOFA score including the acylcholines)	
Fatty Acid metabolites (10/145)	*N*-acetyl-2-aminooctanoate *, palmitoylcholine, dihomo-linolenoyl-choline, linoleoylcholine * stearoylcholine *, arachidonoylcholine, 12-HETE, *N*-stearoylserine *, 3-hydroxy-3-methylglutarate, glycocholate
NUCLEOTIDE (all 7 increased, six associated with the SOFA score)	
Purine Metabolism, (3/20)	Urate, allantoin, *N*6-succinyladenosine
Pyrimidine Metabolism (4/23)	Orotidine, uracil, 3-(3-amino-3-carboxypropyl)uridine *, 5,6-dihydrothymine
COFACTORS AND VITAMINS (6/38; all 6 increased, and 5 were associated with the SOFA score)	
Nicotinate and Nicotinamide Metabolism	*N*1-methyl-2-pyridone-5-carboxamide, *N*1-methyl-4-pyridone-3-carboxamide
Pantothenate and CoA Metabolism	pantoate
Ascorbate and Aldarate Metabolism	ascorbic acid 3-sulfate *, 2-*O*-methylascorbic acid
Vitamin B6 Metabolism	Pyridoxate
ADDITIONAL (25 identified metabolites, two of them increased, 10 associated with the SOFA score)	
Xenobiotics (29/275)	6 food components, 11 drug metabolites, 4 chemicals, 3 bilirubin degradation products, 1 glycine conjugate, 4 unidentified

All individual metabolites showing a significant difference in ANOVA analysis are listed in the figure according to their classification. The table presents the classification and identity. Metabolites showing decreased levels in Sepsis-3 patients are presented in a green color (* The metabolite identity not confirmed based on a standard).

**Table 2 biomolecules-13-00223-t002:** Differences in individual metabolites between patients with and without bacteremia.

Subpathway	Identity
AMINO ACIDS AND PEPTIDES (8 out of 11 metabolites correlated with SOFA	
Lysine Metabolism	*N,N-*dimethyl-5-aminovalerate
Tryptophan Metabolism	Picolinate, 6-bromotryptophan
Methionine, Cysteine, SAM and Taurine Metabolism	Cystathionine, alpha-ketobutyrate
Urea cycle; Arginine and Proline Metabolism	*N,N,N*-trimethyl-alanylproline betaine (TMAP)
Gamma-glutamyl Amino Acid	gamma-glutamylphenylalanine, gamma-glutamyl-2-aminobutyrate
Fibrinogen Cleavage Peptide	fibrinopeptide B (1-11) *, fibrinopeptide B (1-12) *
Modified Peptides	*N,N*-dimethyl-pro-pro
ENERGY (the metabolite was associated with SOFA score)	
TCA Cycle	2-methylcitrate/homocitrate
LIPIDS (11 out of 16 correlated with SOFA score)	
Fatty Acid Synthesis	Malonylcarnitine
Short Chain Fatty Acid	butyrate/isobutyrate (4:0)
Fatty Acid, Dicarboxylate	decadienedioic acid (C10:2-DC) *
Fatty Acid Metabolism (Acyl Carnitine, Monounsaturated)	ximenoylcarnitine (C26:1) *
Fatty Acid Metabolism (Acyl Carnitine, Dicarboxylate)	octadecanedioylcarnitine (C18-DC) *, octadecenedioylcarnitine (C18:1-DC) *
Fatty Acid Metabolism (Acyl Carnitine, Hydroxy)	3-hydroxyoleoylcarnitine
Fatty Acid, Monohydroxy	2-hydroxydecanoate, 3-hydroxyhexanoate
Eicosanoid	leukotriene B4
Progestin Steroids	5alpha-pregnan-3beta, 20alpha-diol disulfate, pregnanediol-3-glucuronide
Corticosteroids	cortisol 21-sulfate
Primary Bile Acid Metabolism	Taurocholate
Secondary Bile Acid Metabolism	taurochenodeoxycholic acid 3-sulfate
NUCLEOTIDES (none of the three metabolites associated with SOFA score)	
Purine Metabolism, (Hypo)Xanthine/Inosine containing	Xanthine
Pyrimidine Metabolism, Uracil containing	5-methyluridine (ribothymidine)
Pyrimidine Metabolism, Thymine containing	3-aminoisobutyrate
COFACTORS AND VITAMINS (one metabolite associated with SOFA score)	
Vitamin A Metabolism	Retinol (vitamin A)
XENOBIOTICS (only 3 out of 19 metabolites associated with SOFA score)	
Food Component/Plant	methyl glucopyranoside (alpha + beta), vanillic acid glycine, 4-vinylguaiacol glucuronide
Drug—Analgesics, Anesthetics	3-(*N*-acetyl-cystein-*S*-yl) acetaminophen, 4-acetamidophenylglucuronide, 2-hydroxyacetaminophen sulfate *, 2-methoxyacetaminophen sulfate *, 2-methoxyacetaminophen glucuronide *, 3-(cystein-*S*-yl)acetaminophen *, 3-(methylthio)acetaminophen sulfate *
Drug—Cardiovascular	Metoprolol, metoprolol acid metabolite *, alpha-hydroxymetoprolol, warfarin, 6-hydroxywarfarin, 7-hydroxywarfarin, 10-hydroxywarfarin
Drug—Gastrointestinal	Pantoprazole
Chemical	2-acrylamidoglycolic acid

All individual metabolites showing a significant difference in ANOVA analysis are listed according to their classification; metabolites with increased levels in patients without bacteremia are marked in red (* The metabolite identity not confirmed based on a standard).

**Table 3 biomolecules-13-00223-t003:** Subclassification of 60 patients fulfilling the Sepsis-2 criteria based on 23 amino acid metabolites that showed a highly significant association with total SOFA score (i.e., Pearson’s test, *p* < 0.005 and correlation factor > 0.60). The continuous variables are presented in the table as the median and variation range. The Fisher’s test was used for comparison of continuous data and the Mann-Whitney U test for analysis of continuous data.

Parameter	Upper Subcluster (*n* = 26)	Lower Subcluster Plus Lower Main Cluster (*n* = 34)	*p*-Value
Number of Sepsis-3 patients	8	27	0.002
Total SOFA score	1 (1–12)	3 (0–16)	0.0074
Number of patients with cardiovascular comorbidity	0	9	0.0038
Age (rears)	54 (20–83)	72 (23–96)	0.0076
Peripheral blood neutrophil count (×10^9^/L)	12.9 (6.4–1.1)	10.6 (2.4–26.6)	0.55
Peripheral blood platelet count (×10^9^/L)	237 (58–568)	176 (24—407)	0.080
Serum creatinine (mmol/L)	66 (27–417)	121 (51–706)	0.0010

We identified a lower main patient cluster including six patients (Figure 1). The upper main patient cluster included the remaining 54 patients who formed two subclusters; the upper subcluster included 26 patients and the lower subcluster included the remaining 28 patients. The table presents a comparison of the upper 26 patients (i.e., the upper subcluster) versus the lower 34 patients (i.e., the 28 patients in the lower subcluster plus the six patients in the lower main cluster). The sample storage time did not differ between the two groups.

**Table 4 biomolecules-13-00223-t004:** Subclassification of the 35 Sepsis-3 patients based on the unsupervised hierarchical clustering analysis presented in Figure 3. The continuous data are presented as the median and variation range. The Fisher’s test was used for comparison of continuous data and the Mann-Whitney U test for analysis of continuous data.

Parameter	Sepsis-3 Patients in the Upper Subcluster (*n* = 8)	Sepsis-3 Patients in the Lower Subcluster Plus Lower Main Cluster (*n* = 27)	*p*-Value
Total SOFA score	3 (2–7)	4 (1–16)	0.32
Number of patients with cardiovascular comorbidity	0	9	0.037
Empty row			
Age (rears)	62 (34–66)	75 (23–96)	0.048
Peripheral blood neutrophil count (×10^9^/L)	10.1 (6.4–41.1)	10.3 (3.4–26.6)	0.55
Peripheral blood platelet count (×10^9^/L)	231 (58.538)	162 (24–407)	0.46
Serum creatinine (mmol/L)	77 (54–107)	130 (51–475)	0.0139

This hierarchical clustering was based on the 23 high-rated amino acid metabolites that showed a highly significant association with the total SOFA score (i.e., Pearson’s test, *p* < 0.005 and correlation factor > 0.60). We identified a lower main cluster including six patients, whereas the upper main patient cluster included the remaining 54 patients who formed an upper subcluster including 26 patients and a lower subcluster including 28 patients (see Figure 3). The table presents a comparison of the eight Sepsis-3 patients among the upper 26 patients (i.e., the eight Sepsis-3 patients included in the upper subcluster of the main patient cluster in Figure 3) versus the 27 Sepsis-3 patients who were included among the other (lower 34 patients in Figure 3 (i.e., the 28 patients in the lower subcluster plus the six patients in the lower main cluster)). The sample storage time did not differ between the two groups.

## Data Availability

The data presented in this study are available on request from the corresponding or first author.

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
