# Peer review of "Systemic Metabolomic Profiles in Adult Patients with Bacterial Sepsis: Characterization of Patient Heterogeneity at the Time of Diagnosis"

_biomolecules, 2023, doi:10.3390/biom13020223_

Round 1

Reviewer 1 Report

the paper can be published, but more informations on the clinical use are important

Reviewer 2 Report

The reader wonders why this paper is based on ten year old patientdata and why no follow-up information is provided.

How can all patients admitted with (severe) sepsis sign an informed consent?

The description of patient recruitment into groups is not clear and should be replaced by a conventional flow chart.

A study of heterogeneity among sepsis patients is described as a main objective. Although this aim is clinically relevant, this is however not possible without a clear description of the source of sepsis, the duration of symptoms, and a detailed reporting of SOFA scores. Provision of follow-up data with repeated scores and survival data would have increased the scientific value of the study significantly.

The metabolomic analyses are well performed and presented, but without a clear description of patient groups the clinical relevance of the study vanishes.

Reviewer 3 Report

Mosevoll et al conducted an interesting study investigating metabolomics in adult sepsis patients. It is well written and encompasses many analyses. I think the authors can improve this manuscript by clearly describing with what goal exactly each of the analyses was performed. For me it now is often not very clear why or with what objective the analysis is performed.  Below my suggestions, questions and comments

Patients general (line 77):

Do the authors have any information on pre-hospital treatment? Can this treatment be different between the groups of the various comparisons (eg sepsi 2 vs sepsis 3) and could they explain differences in metabolites?

Patient selection to 60 patients (line 91): Authors describe how they have come to 60 patients in adequate detail. Did the authors perform any power calculations to assess whether this group size was adequate to answer all research questions they investigated?

2.1 Line 107: 19 of the 80 sesis 3 patients; the number 80 is not compatible with the 35 in line 112.

Patients Line 113 - 131 ->authors describe their cohort here. I would argue these are more appropriately described in the methods section or supplement.

2.3 stat/bioinf line 144: Can imputing missing values with the minimum observed valu for each metabolite lead to problems if the metabolite is downregulated in sepsis? In table 1 the green metabolites are the onses that show decreased levels as is mentioned. Is this the best way of imputing

2.3 Line 157-161: Who did define the pathways that were investigated in the enrichment analysis? 

2.1. patients/methods general: A lot of the analysis look at correlation with SOFA score. I would expect also a section on how SOFA scores are derived (which source) at what time point etc. Is the SOFA at admission? Or highest during hospital stay? Etc.

Result general: Many comparisons are made. When reading I am often puzzled about why specific analysis are performed. What is the goal of that analysis. This could be more clear.

Result general: Consider putting in some of the results in the supplements and focus on the most important analysis for which you started the analysis. Were there prespecified questions? 

Discussion general: can the authors elcudiate on whether underlying pathways could lead to new drug development?

Results 3.3 225-234: Numbers of metabolites are difficult to follow here. 51 meatbolites - 11 and 16 in largest group. Nine (consider to put 9 there to improve readibility?) xenobiotic metabolites lead to 42 remaining metabolites. 10 of them show increased level (rest decresed??) and this is different from 21vs 21 if caused by incidence? What is the purpose of investigating this?

Results general: Some comparisons have another sample size than 60 (eg gram- vs gram +). This should be added in each section.

Result 3.5 line 279-281: this sentence seems to miss some words? Reads difficult.

Results 3.5 line 285: I think the number 34 should be 28 (six, consider 6?, are added in next part of the sentence.

Results 3.6: Perhaps consider to move this to supplement? again; why was this analysis performed. 

Round 2

Reviewer 2 Report

The authors have amended the paper in accordance with the reviewers` comments.

The paper is significantly improved and I am happy to recommend that it is accepted in its present form.